# Misadventures of Sentience: Animals and the Basis of Equality

**DOI:** 10.3390/ani9121044

**Published:** 2019-11-29

**Authors:** Federico Zuolo

**Affiliations:** Department of Classics, Philosophy and History, University of Genova, via Balbi 30, 16126 Genova, Italy; federico.zuolo@unige.it

**Keywords:** basis of equality, equal consideration of interests, equality in animal ethics, sentience, proportionality, Alasdair Cochrane, Peter Singer

## Abstract

**Simple Summary:**

Equal moral worth in animal ethics is an elusive moral notion not only because of the notorious human prejudice but also because grounding equal moral worth requires attending to the problem of the basis of equality. How can we ground equality given that all human and nonhuman individuals vary in all the morally considerable features? John Rawls claimed that we can use range properties, namely properties that are equally possessed by all people who pass a certain threshold of moral relevance (e.g., the age of majority gives equal right to vote). In this paper, I critically discuss two different attempts to defend an egalitarian theory in animal ethics: Alasdair Cochrane’s and Peter Singer’s. The former seeks to eschew the problem of range properties by appealing to a binary property naturally possessed by all sentient beings (the property of having interests). His attempt fails because this property has the same problems as range properties. The latter dispenses with equal moral worth altogether by defending the principle of equal consideration of interests. I argue that this principle has weak egalitarian credentials. I conclude by outlining the conditions that a sound theory in animal ethics should meet.

**Abstract:**

This paper aims to put in question the all-purposes function that sentience has come to play in animal ethics. In particular, I criticize the idea that sentience can provide a sound basis of equality, as has been recently proposed by Alasdair Cochrane. Sentience seems to eschew the standard problems of egalitarian accounts that are based on range properties. By analysing the nature of range properties, I will show that sentience cannot provide such a solution because it is constructed as a *sui generis* range property. After criticizing the approaches seeking to ground animals’ equal status, I turn to Singer’s principle of equal consideration of interests. Despite its seeming non-controversiality, I argue that it cannot do without referring to the moral status of a being in order to determine the weight of a being’s interests. Moreover, it outlines a weak egalitarian basis because it relies on the presumption of equality of interests in virtue of our lack of knowledge of the weight of individuals’ interests. I conclude in a more positive tone by arguing that, irrespective of the troubles of range property egalitarianism, animal ethics can rely on other normative resources to defend the cause of animals.

## 1. Introduction

Sentience is the most popular currency in animal ethics. It is not only employed in academic literature but also appealed to as a morally relevant characteristic in public debates. Sentience has become the hallmark to individuate morally relevant individuals. Part of the success of sentience is due to it being less controversial than other morally relevant properties such as Tom Regan’s being a subject-of-a-life. Moreover, it seems to be uncontroversial in its meaning, and its possession seems easily ascertainable. In other words, it seems to be a “natural” concept with obvious implications that nobody could reject. Another advantage is that it seems to be acceptable by consequentialist as well as deontological perspectives. However, this seemingly universal acceptance might have some unscrutinised issues.

In this paper, I want to challenge the all-purpose function that sentience has played in animal ethics, particularly with respect to its egalitarian credentials. I will argue that it has similar problems as other, less inclusive concepts such as the idea of personhood. I will focus on the idea that sentience can overcome the problems of range properties (personhood and being the subject-of-a-life) that are considered unduly discriminatory. This detour on sentience and the basis of equality will provide us with further considerations concerning the problems of an egalitarian theory in animal ethics and the more general problem of the basis of equality.

The appeal to sentience has been wide and convincing for many. In the market of the properties to grant moral worth, sentience has been used as an alternative to personhood, moral agency, rational capacities, and so on. All these properties that are meant to grant moral worth (and rights) to members of the human species and not to animals falling prey to the standard problem of species overlap: some members of the human species do not possess the required (level of) property necessary to feature as having moral worth, while some nonhuman animals do. This is true of probably all the properties that are typically appealed to when trying to justify the specialty of humans. The argument goes as follows: Switching to sentience to guarantee moral worth is immune to the problem of species overlap because it is a trans-specific feature that is certainly morally relevant and that human beings and many animals share. Hence, the argument concludes, if we want to avoid the problem of speciesism, we should opt for sentience as a ground for moral worth and not for other traditional properties thought to guarantee privileges to human beings, thus granting equally moral worth to all sentient beings.

However, what should we make of sentience? What kind of rights, if any, does it grant? What kind of morally appropriate response does it demand? Once the basic point of claiming that sentience establishes the award of moral worth is made, many diverse solutions are proposed. The first question to be answered is this: Does sentience grant an egalitarian or a non-egalitarian moral response?

To address this problem, I will first clarify what range properties and sentience are. Next, I will discuss the idea that sentience might be a solution to the problem of grounding equality. I will show that these arguments fall short of the mark because the properties that they propose (in particular, sentience) have the same problem as the range property that they attack. Then, I will broaden the focus and touch upon another kind of approach based on sentience—namely Peter Singer’s—that seems immune to these problems insofar as it purportedly abandons any range property. I will argue, however, that Singer’s account in fact abandons a convincing commitment to equality. The implications of this critical analysis will be that egalitarian theories in animal ethics either suffer from the same problems as human-based egalitarianism or are egalitarian only in a very flimsy sense. However, this should not be seen as a defeat for the case of animal interests because other (non-egalitarian) normative resources also come into play.

## 2. What is a Range Property?

Before evaluating the prospects of grounding equality on sentience, we need to discuss the other main element of this analysis: range properties. I will start by analysing Rawls’s take on range properties. Although he did not contribute to animal ethics and others have employed range properties, he coined the term and his analysis is the first thorough reflection on range properties as a basis of equality.

Facing the problem of grounding equality against the background of human differences, Rawls proposed the idea of a range property. Human beings vary in intelligence, capacity for moral agency, conscientiousness, and so on. However, 

It is not the case that founding equality on natural capacities is incompatible with an egalitarian view. All we have to do is select a range property (as I shall say) and give equal justice to those meeting its conditions. For example, the property of being in the interior of the unit circle is a range property of points in the plane. All points inside this circle have this property although their coordinates vary within a certain range. They also equally have this property, since no point interior to a circle is more or less interior to it than any other interior point [1] (p. 508).

Rawls gave particular importance to the two moral powers: the capacity for a sense of justice and for a conception of the good. These are subvenient properties upon which the idea of moral personhood supervenes. Hence, independently of the variations of our sense of justice or capacity for a conception of the good, in Rawls’s view, all people who pass a certain threshold of these capacities should be equally considered persons.

More generally and irrespective of the specific solution proposed by Rawls, the idea of range property can be put as follows. Although morally relevant features vary in degree, we might consider the individuals who possess the relevant capacities within a certain level as equals. In other words, all those individuals who pass a threshold (and are below an upper threshold) should be considered equals because they possess the range property (of being persons, moral agents, and so on) irrespective of whether they possess more or less the underlying morally relevant features. 

This reasoning should not be seen as a self-serving move for it is quite common in many domains of normative reasoning. Consider the case of residence: a person living close to the boundary of a certain nation and one living in the middle of such a nation are thought to be equally resident in such a territory. The same applies, for instance, to the case of voting rights or other citizenship-related goods because they are conferred equally even though people’s capacities (e.g., to vote democratically) vary. The move of establishing a range property upon scalar and variable properties should be used when it is appropriate, namely when we have reasons to treat equally cases that differ in some relevant respects.

Unsurprisingly, the proposal of adopting a range property as a basis of equality has been greeted with a barrage of standard objections such as *arbitrariness of the threshold*, *variability above the threshold*, and *exclusivity*. I draw on the three standard objections outlined by Parr and Slavny [2]. However, I reformulate their third objection by widening its scope beyond humanity in terms of Exclusivity.

*Arbitrariness of the threshold*. The idea of a threshold dividing those who merit moral consideration and/or rights from those who do not has been considered morally unacceptable. There are tiny differences between those who are just below the threshold and those who are just above it, but the treatment they receive is very different and, therefore unfair, and unjustifiable.

*Variability above the threshold*. If what bestows moral considerability or rights is the possession of intelligence or capacity for agency, why should we be egalitarian within the range property? Why should we not track the differences within the range property? After all, it is a commonsensical yet morally relevant fact that people vary in their intelligence and sense of justice and that animals vary in their agency, capacity to cooperate, and intelligence.

*Exclusivity*. Range properties might be seen as troublesome also because the imposition of an equal consideration within a certain class of individuals comes at the cost of excluding other individuals. This, combined with the troubles of establishing the threshold (*arbitrariness*), makes the attribution of range properties an exclusionary affair that grants equality only to a restricted set of individuals.

I do not think that these criticisms are necessarily fatal to range property egalitarianism, but I want to focus on the idea that sentience might be the ultimate morally relevant property that solves the problem of the approach based on range properties. The idea is that sentience can solve these problems because it is binary by its very nature.

## 3. Why Sentience Seems to be a Solution

Before proceeding, we should clarify what sentience is and how this concept is construed in standard accounts of animal ethics. Although the idea of sentience might appear at a first glance uncontroversial, there are many accounts of it. Here, we are not interested in adjudicating between them. It is sufficient to mention some of the properties that are usually appealed to when trying to define this notion. A minimalist definition would simply have it as a matter of feelings; sentience can be conceptualized as the capacity to have feelings. Then, the controversy would turn to explaining what it means to have feelings. A more inclusive and rich account lists a series of capacities: “a sentient being is one that has some ability (i) to evaluate the actions of others in relation to itself and third parties; (ii) to remember some of its own actions and their consequences; (iii) to assess risks and benefits; (iv) to have some feelings; and (v) to have some degree of awareness” [3] (p. 5). In between these two definitions, there can be a mix of diverse options combining a thinner or richer list of capacities.

In animal ethics, the capacity to feel pleasure and pain—which was the first feature bookmarked by Bentham as the fundamental criterion for moral consideration—has been often taken as the shorthand for sentience. Of course, sentience is not only the capacity to feel pleasure and pain, even though such a capacity has been used as a shorthand for sentience by many authors and not only by utilitarians. Irrespective of this, there is the idea that sensory capacities and the type of nervous system generate individual experience. In brief, we might say that sentience entails a certain sensory capacity based upon a sufficiently developed nervous system such that an individual has experience of its interaction with the environment. Without delving into the intricate debate about animal minds, the striking yet straightforward recognition of the similarity between animals and human beings in their being sentient has been the cornerstone of animal ethics. Many have held that, by taking into account the contemporary knowledge in biology, we could draw radical moral implications on animals *qua* sentient beings [4,5]. All diverse approaches in animal ethics have relied on the idea that the scientific evidence provided by the Darwinian revolution constitutes sufficient ground to treat animals in an ethical way, and such a ground hinges on the recognition of sentience. Much more should be said about the different understandings of sentience, their relation to scientific evidence, and their moral implications. However, we can dispense with further details because the following considerations about sentience as a possible binary property are not affected by the variations in the accounts of sentience. 

To analyse the idea that sentience can overcome the problems of range properties, let us consider Alasdair Cochrane’s take on this. Cochrane proposes sentience as the mark of moral considerability. To do this, he relies on the widely held assumption that only sentient beings have interests of their own. Even plants and other non-sentient beings can be benefited or harmed, but only sentient beings can be benefited or harmed as individual beings insofar as they experience as individuals the positive or negative effects of what happens. Hence, what marks equal moral considerability is sentience, which confers upon sentient beings the property of having interests. However, the striking aspect of Cochrane’s proposal is that, in his view, sentience not only is capable of establishing moral considerability (as it does in many other accounts) but also does so in an egalitarian manner. The argument, which is revealing of an overall mistaken tendency, is worth being cited at length:

After all, unlike the capacities for personhood or the characteristics based on cognitive complexity, the possession of interests is “binary.” That is to say, an individual either possesses interests or does not, making it straightforward to explain why the moral worth of humans does not come in degrees. Of course, this is not to say that there are no difficult cases when it comes to identifying which individuals have interests and which do not. […] But still, these uncertainties do not detract from the fact that some individuals are sentient and others are not; and thus that some possess interests, while others do not. Of course, none of this is to deny the obvious fact that interests are differentiated in numerous ways: individuals will often have different *types* and *numbers* of interests; and even when individuals have the same interest, it may vary in *strength* and *complexity*. Nonetheless, it is impossible for any individual to be more or less *in possession of interests*: an individual either has them, or does not [6] (p. 24).

First, it is blatantly false that interest possession should necessarily be understood as a binary property. Second, it is one thing to say that an individual either has or does not have interests but it is quite another to say that worth does not come in degrees. Let us unravel the problems in Cochrane’s argument. One can obviously say that an individual is more in possession of interests than others by way of the idea that being in possession of interests can be made scalar in virtue of interests being (as Cochrane says) more or less complex, more or less strong, etc. Indeed, one can adopt either a scalar or a range understanding of such a property depending on the kind of analysis one does. This is possible because having interests is a property that supervenes on the strength and complexity of certain subvenient properties that are subsumed under the idea of having interests. I contend that there is a hidden contradiction in Cochrane’s argument; the construction of having interests as a binary property mirrors the construction of personhood from which Cochrane wanted to depart.

Suppose we consider the capacity of having interests as a binary property, as Cochrane urges. We can conceive of this capacity in many ways depending upon how we understand it as composed of a varying set of subvenient properties. For instance, such a capacity can be determined by the capacity to feel pain and to have memory, and more generally, it depends on the kind of central nervous system an individual has. All these properties can be understood as scalar properties. This means that either a certain level (threshold) of the subvenient capacity is sufficient to trigger the full possession of the range property or, more likely, a combination of certain levels of diverse capacities triggers the attribution of the capacity of having interests as a range property. In any case, the construction of interest possession is analogous to the construction of personhood—the capacity that Cochrane criticizes as a range property in Rawls’s account—while they differ in the subvenient capacities, the normative implications, and the range of its possessors. This means that sentience has the same kinds of problems as personhood in terms of arbitrariness of the establishment of threshold, lack of proportionality in the treatment that two individuals just above and just below deserve, and so on.

This trouble is not confined to Cochrane’s theory. A similar problem is encountered by Rainer Ebert’s proposal. He rejects egalitarian range properties, particularly McMahan’s three-tiered account and Regan’s theory, with some of the arguments we have seen. He is, in particular, dissatisfied with the arbitrariness of setting a threshold that inevitably separates similar individuals and establishes huge differences between them. So far, so good. Against this background, he proposes an egalitarian moral status to be accorded to all conscious beings:

The line that separates somebodies from somethings, conscious from non-conscious life, is anything but arbitrary and seems to signify a remarkable distinction in nature. While I suppose one could take a gradualist view about consciousness and its emergence in both evolution and the development of individual animals, the competing view that being phenomenally conscious is a strictly binary property has at least equally as much initial intuitive plausibility. For each animal, either there is something it is like to be that animal, or not [7] (p. 15).

Ebert seems aware that his move looks very much like the one he is criticizing, but he is confident that the chosen property could have better credentials than the others because it is a natural property that should come in a binary manner. As seen in the case of Cochrane, the usual problems of establishing a threshold and of constructing a range property supervening on variable subvening properties emerge here. In sum, Cochrane and Ebert search for a “natural” binary property that eschews the problems of range properties, but their attempts fail.

I have not argued here that, in general, there are no empirical properties that are possessed per se in a binary way, such that the possession of these properties would meaningfully justify the attribution of moral status or rights equally. Rather, I am just arguing that the attempts above have the same problems as the standard accounts that explicitly employ range properties and/or attribute range properties only to humans.

It is worth noting again that range properties have been famously used in animal ethics before the emergence of these accounts that purportedly seek to avoid range properties. The most important of them is Tom Regan’s idea of *being a subject-of-a-life*. On Regan’s view, all mammals aged one or more years have memories, perception, expectations of the future, and self-consciousness that make them individuals whose mental complexity is similar to that of standard human beings [8] (p. 81). For these reasons, all those who pass the threshold are subjects, namely individualities that are more than mere sentient beings. On this basis, all subjects-of-a-life should be granted equal moral worth and rights.

Unlike Cochrane, Regan does not seek to shy away from the limits of range properties—but his theory has a serious drawback. If we compare Regan’s theory with other accounts in animal ethics, it is immediately apparent that the criterion of being a subject-of-a-life does not apply to most animals. Accordingly, we have an egalitarian theory with an extremely restricted scope. In fact, Regan decided to extend the treatment owed to the subjects-of-a-life to many other animals as a matter of moral caution [8] (p. 366). However, this move is dubious for his argument in principle has a more restricted reach. Against this backdrop, we might think that the prospects of an egalitarian theory in animal ethics are bleak because the attempts to ground equal status are either too limited in their reach (Regan) or they fail to properly acknowledge and justify the adopted range property (Cochrane). However, this is not necessarily the case because a theory might seek to be egalitarian without endorsing equality of status, as we will see in the next section.

## 4. Equality and the Principle of Equal Consideration of Interests

Against this backdrop, how does the “champion” of sentience as a moral criterion—Peter Singer—fare with respect to the prospects of an egalitarian account in animal ethics? This is not a futile question because the problem of how to ground equal moral worth despite the variation of features has been called the “Singer problem” [9]. Indeed, Singer has famously formulated an important critique of the idea of moral status and Rawlsian range property. First, he rejects the idea that moral personality can be attributed as a property equally possessed by diverse humans. Indeed, “having moral personality is a matter of degree” [10] (p. 18). Second, he claims that many humans cannot be said to have moral personality insofar as they have permanent, temporary, or acquired deficits. The most straightforward implication from these two points would be to adopt a hierarchical approach in which rights and worth are assigned on the basis of the varying possession of morally relevant properties. However, Singer rejects this “hierarchy of intelligence” because “the claim to equality does not rest on the possession of intelligence, moral personality, rationality, or similar matters of fact. There is no logically compelling reason for assuming that a difference in ability between two people justifies any difference in the amount of consideration we give to their interests” [10] (pp. 20–21). To solve this problem, Singer abandons the idea of equality of status but sticks to the idea of relocating some form of equality at another site.

Equality, indeed, should be accorded to interests. The famous principle of equal consideration of interests requires that “we give equal weight in our moral deliberations to the like interests of all those affected by our actions” [10] (p. 21). Despite the seeming clarity and straightforwardness of this principle, one might ask what is egalitarian in this principle. Is it not obvious that like interests should be given equal weight? Singer wants to prevent unequal treatment of individuals who differ in their species belonging but have a similar kind of interest at stake.

We are used to considering this principle as the hallmark of egalitarianism, but perhaps this is not so obvious. Digging deeper into this principle might reveal its ambiguity. Shelly Kagan has argued that this principle cannot exclude that the moral status of a being is relevant in the determination of the interest at stake because moral status is a factor that affects the kind of well-being of an individual [11] (pp. 101–118). Although I agree with Kagan’s point, here, I want to tackle a different aspect, namely its egalitarian credentials. In what sense is this principle egalitarian? This is not a specious question because, after defining it, Singer puts in question the egalitarian pedigree of his principle by turning back to a coherent proportional determination of one’s interests by considering the case of pain:

Then the principle says that the ultimate moral reason for relieving pain is simply the undesirability of pain as such, and not the undesirability of X’s pain, which might be different from the undesirability of Y’s pain. Of course, X’s pain might be more undesirable than Y’s pain because it is more painful, and then the principle of equal consideration would give greater weight to the relief of X’s pain. Again, even where the pains are equal, other factors might be relevant, especially if others are affected [10] (p. 21).

The reasoning here could be as follows. First, we should establish whether an individual is sentient and, if so, to what extent (bear in mind that, in Singer’s view, there is a difference between merely conscious and self-conscious individuals). Second, we should check whether such an individual has the interest at hand. Third, we should weigh the importance of such an interest. Fourth, we should compare it to the weight of other interests that are possibly competing in a certain situation—assuming that, if there is no competition, meeting the interest at stake should have no countervailing considerations. None of these passages express an egalitarian commitment. In the first passage, one should possess a certain capacity, which seems a binary assessment. However, nothing prevents a utilitarian like Singer from saying that the possession of a capacity is unequally possessed, as he often and correctly does. The second passage merely requires a factual judgment to ascertain a matter of fact. The third passage can consider only differences, and the fourth aims at comparing and weighing the relative importance of interests. Thus, where does its egalitarian reputation come from? There is perhaps only one sense in which Singer’s principle of equal consideration of interests is egalitarian. It is a minimal sense of egalitarianism in terms of nondiscrimination:

The principle of equal consideration of interests acts like a pair of scales, weighing interests impartially. True scales favour the side where the interest is stronger or where several interests combine to outweigh a smaller number of similar interests, but they take no account of whose interest they are weighing [10] (p. 22).

The principle demands us not to discriminate the interest of a being, only because such a being is supposedly less intelligent, developed, and so on. Rather, it requires us to consider only the weight of such an interest. Hence, this principle demands not to discriminate individuals. What we should do in case of competing interests depends on the assessment of the relative weight and similarity of the interests at stake. Indeed, much of the ambiguity of Singer’s principle rests on the central phrasing that one should give “equal weight” to the “like interests.” What does *like* mean here? Is Singer saying that we should give equal weight to the interest in not suffering of all individuals who are capable of suffering? This is a possible interpretation and the one that Singer himself suggests. However, we can give equal weight to my interest in not suffering and to the interest in not suffering of another human being or sentient animal only *if and to the extent that* one can presume that there is an underlying equal basis for giving equal weight. One should not count more (or less) the weight of an interest only because it is the *same kind* of interest. This would go against one of the most important consequentialist tenets, which are so dear to Singer.

Hence, in this consequentialist account, we should grant equal consideration of interests in two cases: if the two competing interests are of equal weight or if we lack any reason to doubt that the two interests are of equal weight. Call the former the *equality by coincidence* case and the latter the *equality by ignorance* case. In other words, treating two interests as having equal weight either seems a mere coincidence because these two interests de facto have equal weight or seems to be justified only because we do not know how much they actually weigh. I am not claiming here, as Peter Westen [12] did, that formal equality (as he put it, people who are alike should be treated alike) is a mere tautology. I am claiming that if it turns out that two interests are really the same in all respects (weight, duration, intensity, importance, etc.) then it is obvious that they should be considered equally. But the difficulty revolves around the determination of what it means for two competing interests to be equal.

For equality by coincidence to be obtained, we must have an “interestometer” that measures the weight of competing interests. Absent of an “interestometer,” we are left with equality by ignorance. In this sense, it is in virtue of our epistemic limitation that we should give the benefit of doubt and grant equal consideration to all cases in which two presumably similar interests are conflicting. This is not easy to swallow for Singer because his approach is notoriously squeamish about epistemic limitations and would rather urge us to look for more knowledge so as to have a more fine-tuned judgment of the relative weight of the two interests. It would be an unwelcome conclusion if the egalitarian flavour in Singer’s account depended on our epistemic and technological limitations.

Perhaps this is too hasty. The principle of equal consideration of interests works as a negative tool that urges us not to discriminate. It entails a commitment to treating equally, when it is the case, without being biased by irrelevant factors (sex, race, intelligence, and species). In principle, the egalitarian treatment should be determined by the equality of the weight of the interests at stake. Thus, the weight of the interest is the basis of the normative consideration. Here, we face a problem of which the solution depends on the approach in value theory we choose—is the value of interests determined by one’s subjective preferences, objective preferences, or the intrinsic value of states of affairs? If we were in the business of establishing whether two specific interests weigh equally, we should choose a suitable value theory and apply it to the case at hand. Whatever the solution in value theory, the important thing is that the principle of equal consideration of interests simply tells us to focus on the interests—whatever its value basis—and to disregard the individuals that have such interests. However, the case is different if we just focus on the interest and take the principle as fully valid, as its egalitarian credentials are flimsy.

Suppose we want to apply the principle of equal consideration of interests and we have two competing interests but do not know from whom they come. Suppose also that we only know the two interests are equal because an “interestometer” told us so. In that case, the principle of equal consideration of interests comes as an obvious and compelling response. However, once the interests are—as it were—detached from their holders, the egalitarian commitment is merely tautological if we know the interests are really equal in weight. After all, if we only know that two competing interests demand *x* and have the same weight, why should we treat them differently? If, instead, we have before us two diverse individuals (whether human or nonhuman) who, suppose, have a competing interest in being fed, how could we consider their interests on a par without scrutinizing the kind of individuals they are? Is my interest in being fed on par with a snake’s interest?

Supporters of the principle of equal consideration of interests are likely to answer affirmatively but to lessen the impact of the implications that follow. Indeed, the typical answer in animal ethics and the one adopted by both consequentialist and deontological accounts puts in place a manoeuvre to bypass the question when facing the comparison of values of different lives. By appealing to the fact that a person has more valuable experiences than that of a nonhuman animal, such accounts aim to hold on to the equality principle while allow a different treatment between human beings and nonhuman animals. They seem committed to the idea that the same interest—say, in being fed or in not suffering—has the same weight, notwithstanding the fact that persons also have other interests. Although I do not think this manoeuvre is necessarily wrong, it gives the impression of moving the site where the comparison of value is made (from interests to lives). This point is affirmed in a more critical and categorical way by John Rossi [13]. He argues that equal consideration of interests is incompatible with unequal moral status and that theorists in animal ethics should either drop the principle of equal consideration or abandon unequal status. Indeed, the idea of allowing unequal status is functionally similar to what I have just described as a matter of lives that include smaller or bigger amount of valuable experiences.

At this point, it is worth recalling that David DeGrazia has argued that an egalitarian approach to the comparative value of lives and interests can be justified by the lack of full justification of the alternatives [14] (pp. 248–256). DeGrazia here is addressing the comparison of the values of lives, but this argument is relevant because he needs it to make sense of the equal consideration of interests. In an egalitarian approach, he claims, “it is not argued that we have compelling reasons to assert that the lives of normal humans, monkeys, bluejays, and lizards are equal in value. Rather, it is claimed that *we should regard them as equal*—for lack of good reason to regard them as unequal” [14] (p. 248). This indirect justification of equal consideration is not the same as the point we are considering here about Singer, but it points to a common difficulty for accounts that rely on the principle of equal consideration of interests. Given the impossibility of establishing whether the life of a human being is superior, in terms of the value of one’s subjective experiences, to the life of a dog because the dog has kinds of experiences of which the value is not understandable by us, we ought to regard the value of these lives as equal. The merit of DeGrazia’s considerations is that they expose the epistemic problem that is hidden in Singer’s principle of equal consideration of interests. Against DeGrazia’s conclusion, equality as a default position is not the most epistemically justified position. Indeed, the lack of a common value perspective to handle these comparisons does not justify giving equal weight; rather, it justifies epistemic abstention.

The more general question is whether the assessment of an interest (without an interestometer) can be merely made on the kind of interest itself without considering how an interest relates to the other interests of an individual. Can an interest be gauged in isolation, as if it were a freestanding source of value or disvalue? I doubt that it can because we need to know how an interest is related to other interests as well as the needs and prospects of one’s life. In short, an individual’s moral status serves the purpose of conveying a synthetic summary of these relations. Establishing the implications of this point would need more space, but for the purposes of this paper, these considerations should be sufficient to cast doubts on the supposed egalitarian credentials of the principle of equal consideration of interests.

In sum, the application of the principle of equal consideration of interests rests on there being a mere coincidence of weight in two competing interests or on our presuming that the default is egalitarian consideration because of our lack of means for scrutinizing the value of interests. Either way, this is a flimsy prospect for an egalitarian account.

## 5. Types of Properties and Equality

At this point, it is worth exploring the issue at a greater level of generality because some of the troubles we have encountered stem from a misunderstanding of some conceptual issues linked to equality and the type of properties to which equality is attributed. In particular, it is worth exploring the relation between binary properties and equality.

There are some properties of which possession is necessarily binary. For instance, think of the property of “being a number”. An entity can either be a number or not. There are other properties of which possession is necessarily scalar. Think of the property of being bald. Finally, there are some properties of which possession can be both binary and scalar. Think, for instance, of the property of “having money.” We can meaningfully and appropriately ask whether someone has money or no money at all, and we can also ask how much money one has. By contrast, it makes no sense to ask whether something is more or less a number. We can use many properties as both binary and scalar depending on the pragmatic or conceptual need.

Coming back to our issues, we can say that the property of being sentient can be used both in a scalar and binary manner depending upon what we want to do. Indeed, it makes sense to ask whether a certain entity is sentient or not—to compare, for instance, a stone and a mammal—as well as to ask which entity is more or less sentient. This is because sentience is a property that supervenes upon other physical properties that come in degrees such as the capacity to feel pain, complexity of the nervous system, etc. Consider a very simple animal that is thought to be close to the threshold of sentience, such as a shrimp, and think of the possible kinds of experiences that such an animal can have (temperature, movement, and so on). Although we can hardly be sure of what happens in a shrimp’s nervous system—and I do not want to pose the question of what it is like to be a shrimp—we can suppose that the types and extent of feelings that they can have—provided that they can have feelings—are significantly fewer than those of a mammal in any sense of “fewer”, namely in terms of being fewer in kind, number, complexity, or else. We probably cannot put on a single scale all the features that add up to determine sentience, as if there was only a homogeneous set of features that make sentience. Nevertheless, it is possible (and it makes sense to say) that a shrimp has a less complex nervous system than a mammal, and because of this, the kinds of experience that it can have—if any—are fewer and less complex than those of a mammal. Therefore, although it is a somewhat rough statement, we can safely say that a shrimp is less sentient than a mammal.

The implication of this argument is that those who reject range property egalitarianism often fail by their own standards because they use a range-property-based argument at another level (Cochrane) or end up with a very weak form of equality (Singer). The criticisms we have seen above (arbitrariness, variability, and exclusivity) pose real problems for all moral theories that address the issue of basic equality, whether in a dignitarian, sentientist, or other capacity-based account. However, the problem is more compelling for animal ethicists because they take it almost for granted that exposing the tension between the fact of scalar properties and the issue of the basis of equality is fatal to standard theories grounding the moral worth of humans (and not animals). Humanist theories have problems of their own, particularly with respect to the convincingness of the basis of equality, but the animal rights-based critiques seem unaware that they are prone to the same kind of problem, as they simply switch the level of egalitarian consideration without acknowledging, as other egalitarian theories do, the problem of the basis of equality.

Whatever the prospects of range property egalitarianism in general, all these worries about equality concerning animals and range properties stem from a misguided approach. The debate in animal ethics until a few years ago revolved around the inclusion of animals in the moral domain, which was standardly composed only by human beings. Such a domain was also traditionally characterized by some principle of equal consideration. Then, animal ethicists had to overcome the two joint principles that only human beings deserve moral consideration and that human beings deserve it equally. It is perhaps for this reason that the challenge of overcoming this divide has usually been accompanied by a challenge to apply the principle of equality to animals in some sense. Once animals have been included in the moral realm and have been granted some moral considerability or rights, the battle over equality can be handled more carefully and separately from the more basic one, which concerns the moral considerability of animals. Given the confusion and display of rhetorical engagement in contemporary debates, it is worth noting again the following: it is one thing to say that a set of beings deserve moral consideration, but it is quite another to say that two (or more) sets of beings should be considered equally. This is especially the case because equality cannot be disjointed from the principle of proportionality, as we will see in the next section.

## 6. Equality, Proportionality, and Beyond

Why should we keep equality and proportionality together, given that equality and proportionality are such divergent principles? Indeed, range properties are often invoked to counter the inegalitarian effect of variability of morally relevant properties that might seem to require a proportional principle. A full answer would require a longer detour, but for our purposes, it is sufficient to note that both egalitarian and non-egalitarian principles rely on the overarching Aristotelian principle that equals are to be treated equally and unequals are to be treated unequally (*Nicomachean Ethics* V, 3, 1131 a21–24). Ian Carter nicely summarizes the whole argument:

Egalitarian principles (as defined above) are fact-dependent in the following sense: such principles depend for their normative validity on the fact that individuals are equal in some sense, together with a higher order hypothetical prescription of the form: “If individuals are equal in such-and-such a sense, treat them equally in such-and-such a sense.” This hypothetical prescription is the source of the moral relevance of the fact, and itself derives in part from the still-higher-order Aristotelian principle enjoining us to treat equals equally (and unequals unequally) together with normative claims about the relevance of certain kinds of facts to certain ways of treating people [15] (p. 25).

In sum, there is a tight and yet overlooked relation between equality and proportionality. Since, here, we cannot see how we should handle equality and proportionality in general, I will only seek to frame the matter by providing some considerations to the effect of showing what kinds of errors egalitarian approaches should avoid.

The first is the mistake of adopting a range property that covers an extremely wide range of individuals such that the connection between the range property and the egalitarian principle is too stretched. Paul W. Taylor has argued that all living entities have equal moral status (inherent worth) insofar as all living entities are self-reproducing teleological centres. “Subscribing to the principle of species-impartiality, we now see, means regarding every entity that has a good of its own as possessing inherent worth—the *same* inherent worth, since none is superior to another” [16] (p. 155). This means that all living entities have intrinsic worth because they can be harmed or benefited. In Taylor’s view, this should ground our concern for nature in a way that overcomes an anthropocentric outlook. However, this range property—being a self-reproducing teleological centre—is possessed in such a different way by all living entities that the connection between the range property and the normative implications gets lost. Indeed, what we owe to different entities either depends on the kind of beings they are, thus betraying the equal basis we all share, or (if faithful to the basic range property) it entails such a minimal normative implication—merely considering that every living entity has some inherent worth—that it amounts to assigning very little moral import to inherent worth, as it has been claimed by Richard Arneson. “Either the proposed basis will turn out to vary by degree, and variations above the claimed threshold that establishes equality will give rise to inequality of moral considerability, or the proposed basis will turn out to be one that applies in all-or-nothing fashion, and then it will turn out that the basis proposed as justifying equal moral considerability is too flimsy or insubstantial to do this justifying work” [17] (p. 42). In brief, proportionality gets lost either as an input or as an output. It gets lost as an input when the range property of being a self-reproducing teleological centre grounds equality from algae to human beings because such a range property means very different things across the span of individuals to which it applies; it gets lost as an output when the range property is understood in a very thin sense, thus yielding very minimal normative import. In both cases, one might question why this account should be considered egalitarian; an emphasis on inclusion stretches equality and discards proportionality.

On the other side, a radical proportional account has been outlined by Jeff McMahan in his *Intrinsic Potential Account* [18]. (For another attempt to outline a proportional and coherently inegalitarian theory, see also Knapp [19]). Although McMahan supports an egalitarian range property account for persons, his overall approach is proportional because he holds that the assessment of an individual’s possible misfortune should be made with respect to the possible level of welfare that such an individual could achieve given their individual potentiality. Such a potentiality is determined by one’s level of psychological capacities. This means that the comparative values of individual lives are inextricably variable given one’s varying psychological properties and that it should be so because the ultimate axiological source of value is one’s level of welfare. In this account, equal consideration should be given only to the classes of individuals having the same potential of well-being, which means ultimately the same level of psychological capacities. It follows that egalitarian consideration would apply only to very restricted classes. This seems to be the most inegalitarian version of the Aristotelian principle.

These considerations on Taylor and McMahan are admittedly very sketchy. Nevertheless, they show us that an egalitarian theory recognizing equal status should have a plausible domain of application. If such a domain is too wide, the basis of equality is too thin and then equality loses its normative grip; if it is too narrow, it might lead to inegalitarianism. Which kind of domain is the good one cannot be determined in advance.

The implication of the previous (mostly critical) arguments is that grounding equal consideration for animals is especially impervious. This is not only because of the traditional humanist prejudice but also because of the intrinsic difficulty in grounding equality, whether for a restricted class of beings (persons) or for a large one (sentient beings). Suggestive as it might seem as a prospect of liberation, equality has a big burden of proof, which is sometimes contested even among persons. However, this need not be seen as a despairing state of affairs. There are other avenues to recognize animals’ interest besides claiming for equality. Without defending any of the following proposals as the best one, we can at least mention sufficientarianism and differentiated rights as plausible approaches to defending animal interests given the troubles encountered in finding a basis of equality.

First, one might argue that what matters is that a being leads a sufficiently acceptable life. Since equalization in terms of opportunities, outcomes, or resources is impossible or inappropriate, we should grant each sentient being a sufficient consideration such that it would be enough to live an acceptable life according to its own standard.

An alternative to sufficientarianism is in terms of inegalitarian differentiated rights as outlined by Kagan [11]. Given the extremely varied morally relevant features displayed in the animal kingdom, we might attach diverse levels of moral weight to rights depending not only on the kind of interest that the right defends but also on the moral status of the individual at stake. This perspective does not collapse into pure consequentialism because the moral claims that are expressed through rights are not tradable, albeit they can be counterbalanced by other considerations. Hence, there can be differentiated and unequal rights reflecting the level of moral status of an animal while still being rights in the proper sense.

We cannot discuss the merits and limits of these perspective. What I want to point out is that they do not imply structurally discounting animal interests as somebody might fear. Moreover, these two perspectives have the advantage of taking an animal’s interest for what it is without trying to cover the animal’s moral considerability with too stretched or ungrounded principles of equality.

## 7. Conclusions

To conclude, egalitarianism in animal ethics faces similar challenges as those faced by egalitarian accounts in other domains. Therefore, the idea that sentience might solve these problems, as well as others, is an illusion. On one hand, accounts seeking the justification of equal moral status (Cochrane) eschew the problems of range properties but turn out to be manoeuvres that conceal a use of range properties in disguise. On the other hand, accounts dismissing equality of status (Singer) prove to have a very weak egalitarian pedigree. Although the problems of range properties might not be insurmountable, egalitarian theories in animal ethics seem to lose grip with their basis. Since grounding a meaningful equal consideration for animals is especially difficult, animal ethicists should look for other normative resources to properly defend animal interests.

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
