# Peer review of "Misadventures of Sentience: Animals and the Basis of Equality"

_animals, 2019, doi:10.3390/ani9121044_

Round 1

Reviewer 1 Report

The article is very well written and arguments are exposed in a very clear manner. The topic is absolutely relevant for the debate in animal ethics and the conclusions reached by the author are original and able to substantially contribute to the debate. No major revisions are required and the article is suitable for publication. My suggestion is to slightly improve references to give readers a more detailed scenario of the scientific and theoretical background of the topic discussed in the article. In particular, a few lines could be added when presenting the concept of animal sentience in order to better explain it and  more references should be provided (in particular to scientific literature about sentience). 

Author Response

I'm grateful to the reviewer for pointing out a weakness in my reconstruction of the idea of sentience. I've added some considerations and references at lines 148-163. Of course, much more should be said, but given the word-limit I could not provide more details to this fundamental problem.

Reviewer 2 Report

This is a cutting edge contribution to a central issue in animal ethics/animal politics. It discusses some of the leading approaches grounding the moral standing of animals, theories of a utilitarian or non-utilitarian bent. All build on sentience and promise to honour (some) egalitarian constraints. The paper gives a very clear introduction to its central concepts (range property, sentience) and engages with the most recent developments in the literature. The supervenience/subvenience argument about range properties and natural properties is very acute. The destructive part is successful not only in questioning the false obviousness of the egalitarian entailments of sentience, but also in justifying doubts as to whether sentience is a range property at all. A constructive coherentist element is added to the hypothetical equality of interests, in asking us to embed interests presumptively deserving of equal consideration into the larger context of "the other interests of an individual" (366). But the main innovation of the paper, and its main service for the current debate, is its general reflection on binary and scalar properties, arguing that some properties can be both, which will put future discussion on a new basis, especially as regards egalitarian conclusions. The final section 6 is more explorative than the rest, introduces the difficult topic of proportionality and touches on alternative debates and solutions, but well worth having in this paper since it puts the topic of range properties into a wider context, and at least indicates how to think of alternatives to egalitarianism. In conclusion, I rate the quality and originality of the paper as very high. Its arguments are carefully formulated. I recommend including the paper in the Special issue of Animals as a well-argued innovative addition to the literature on an especially thorny Problem in Animal Ethics. 

Some small points: l. 48 "species-belonging" should be deleted since it is not vulnerable to species overlap. L.88 "no" not "not". L. 148 the step from sentience to interest (in Cochrane) should be explained more fully.L. 166 "First" contains two criticism, the binary nature and the grounding of equality. The second should be discussed separately ("Third") or dropped here. 255 Shelly.

Author Response

I'm grateful to the reviewer for the very careful reading and for pointing out some mistakes. I've edited the document in all the passages mentioned by the reviewer. The clarification of the passage from sentience to interests in Cochrane now is at lines 166-169.